

# Sustained attention in skilled and novice martial arts athletes: a study of event-related potentials and current sources

Javier Sanchez-Lopez[1], Juan Silva-Pereyra[2] and Thalia Fernandez[1]

[1] Departamento de Neurobiologia Conductual y Cognitiva, Instituto de Neurobiologia, Universidad Nacional Autonoma de Mexico, Juriquilla, Queretaro, Mexico
[2] Unidad de Investigacion Interdisciplinaria en Ciencias de la Salud y la Educacion, Facultad de Estudios Superiores Iztacala, Universidad Nacional Autonoma de Mexico, Tlalnepantla, Estado de Mexico, Mexico

Corresponding author
Thalia Fernandez,
thaliafh@yahoo.com.mx

## ABSTRACT

**Background.** Research on sports has revealed that behavioral responses and event-related brain potentials (ERP) are better in expert than in novice athletes for sport-related tasks. Focused attention is essential for optimal athletic performance across different sports but mainly in combat disciplines. During combat, long periods of focused attention (i.e., sustained attention) are required for a good performance. Few investigations have reported effects of expertise on brain electrical activity and its neural generators during sport-unrelated attention tasks. The aim of the present study was to assess the effect of expertise (i.e., skilled and novice martial arts athletes) analyzing the ERP during a sustained attention task (Continuous Performance Task; CPT) and the cortical three-dimensional distribution of current density, using the sLORETA technique. **Methods.** CPT consisted in an oddball-type paradigm presentation of five stimuli (different pointing arrows) where only one of them (an arrow pointing up right) required a motor response (i.e., target). CPT was administered to skilled and novice martial arts athletes while EEG were recorded. Amplitude ERP data from target and non-target stimuli were compared between groups. Subsequently, current source analysis for each ERP component was performed on each subject. sLORETA images were compared by condition and group using Statistical Non-Parametric Mapping analysis. **Results.** Skilled athletes showed significant amplitude differences between target and non-target conditions in early ERP components (P100 and P200) as opposed to the novice group; however, skilled athletes showed no significant effect of condition in N200 but novices did show a significant effect. Current source analysis showed greater differences in activations in skilled compared with novice athletes between conditions in the frontal (mainly in the Superior Frontal Gyrus and Medial Frontal Gyrus) and limbic (mainly in the Anterior Cingulate Gyrus) lobes. **Discussion.** These results are supported by previous findings regarding activation of neural structures that underlie sustained attention. Our findings may indicate a better-controlled attention in skilled athletes, which suggests that expertise can improve effectiveness in allocation of attentional resources during the first stages of cognitive processing during combat.

## INTRODUCTION

Sports performance and training encompass the development of physical, technical-tactical, and psychological skills. Among the psychological abilities, sport training enhances emotional and cognitive aspects. Cognitive processes are essential for optimal sports performance, and attention-related processes are particularly important in combat sports (*Anshel & Payne, 2006*; *Blumenstaein, Bar-Eli & Tenenbaum, 2002*; *del-Monte, 2005*; *Lavalle et al., 2004*; *Rushall, 2006*; *Sánchez-López et al., 2013*; *Sánchez-López et al., 2014*). Previous studies have reported the outstanding attentional capacities of sport experts, who can also more quickly extract and identify the most important and relevant information (*Abernethy & Russell, 1987*; *del-Monte, 2005*; *Sánchez-López et al., 2014*; *Williams & Grant, 1999*). Thus, skilled athletes can better modulate their attention resources according to specific environmental requirements (*Nougier & Rossi, 1999*).

Integrative mind-body training, such as meditation, martial arts, and yoga, is known to enhance brain and cognitive functions, specifically attentional processes (*Brefczynski-Lewis et al., 2007*; *Tang & Posner, 2009*). Focused attention is essential for open-skill sports such as team sports and combat. Since, in combat sports, long periods of focused attention are required during competition, it could be one of the most relevant processes for high performance, and one movement attended or missed can lead to victory or failure, respectively; however, it remains unclear whether a kind of attention related to maintaining focus (i.e., sustained attention) is the key to the performance of experts in these disciplines. Based on this idea, the aim of this study was to evaluate sustained attention in martial arts disciplines.

One method of understanding how sports performance is enhanced is by studying brain electrical activity through the event-related potentials (ERP) technique (*Thompson et al., 2008*). ERP, which are regarded as temporal correlates of information processing (*Jennings & Coles, 1991*; *Picton et al., 2000*), allow us to understand the temporal dynamics of the different sub-processes of a global cognitive aspect such as attention. However, given that attention models involve several brain areas that interact in different ways with every attention subprocess, current source analysis is necessary. The standardized low-resolution brain electromagnetic tomography (sLORETA) is a suitable method to precisely locate brain electrical source.

The role of expertise and training in attention and brain activity has been investigated using the ERP technique and sLORETA. In previous studies (*Babiloni et al., 2010a*; *Del Percio et al., 2010*; *Fontani & Lodi, 2002*; *Fontani et al., 2006*; *Fontani et al., 1999*; *Hack, Memmert & Rupp, 2009*; *Hamon & Seri, 1989*; *Hung et al., 2004*; *Radlo et al., 2001*), behavioral performance, electrophysiological brain activity, and current sources were shown to have distinct characteristics when compared between experts and non-experts or non-athletes, which suggests that people with training in different skills and sports may have attentional profiles related to their expertise. Specifically, ERP studies have found

larger amplitudes in components associated with attention (e.g., P100, P200, and P300) in expert athletes than in other populations (*Hack, Memmert & Rupp, 2009*; *Hamon & Seri, 1989*; *Hung et al., 2004*; *Jin et al., 2011*; *Ozmerdivenli et al., 2005*; *Zwierko et al., 2011*); the authors interpreted these higher amplitudes as indicators of better attentional mechanisms in experts. Few studies have performed current source analysis to investigate cognitive differences between expert and novice athletes. By using sLORETA, *Del Percio et al. (2010)* studied differences in activation of the premotor and motor brain areas during hand movements between karate athletes and non-athletes. Their results showed less activation of these structures in athletes as compared to activation in non-athletes. *Babiloni et al. (2010b)* found differences in the activation of the dorsal and frontoparietal "mirror" pathways between expert, non-expert athletes and non-athletes. Both studies supported the "Neural Efficiency" hypothesis. This hypothesis proposes that efficiency would be observed as spatial cortical reduction of the task-related brain activity in expert athletes when compared with less expert groups (*Babiloni et al., 2009*). However, these studies have not investigated the effect of expertise by analyzing the current sources of electrical brain activity in athletes during any attentional task.

Several ERP studies have investigated the neural correlates of sustained attention by using the continuous performance task and their results have shown different waves that mirror brain electrical modulations to specific demands of attention; among the main ERP components reported in the literature, P100, N100, P200, N200 and P300 are found (for review, see *Riccio et al., 2002*). Although there are no studies that specifically report what areas in the brain are related to sustained attention in athletes, evidence from lesion and functional imaging studies shows that some brain areas, such as the anterior cingulate and dorsolateral prefrontal as well as parietal cortical regions primarily in the right hemisphere, are consistently activated in participants who performed sustained attention tasks (*Cohen et al., 1992*; *Fink et al., 1997*; *Pardo, Fox & Raichle, 1991*). Previous studies using neuroimaging techniques describe sustained attention as a top-down mechanism that begins with the motor and cognitive readiness for the subject to detect and discriminate the stimulus information; this process is mediated by right fronto-parietal brain areas, and it facilitates perceptual and spatial attentional processes that contribute to the performance by recruiting parietal areas related to sensory processing (*Hopfinger, Buonocore & Mangun, 2000*; *Lane, Chua & Dolan, 1999*). The perceptual facilitation of attentional processes via top-down mechanisms could result in increased firing activity in neurons that response selectively in sensory-association areas when attentional task demands are increased (*Desimone, 1996*).

Thus, in the present research we studied the type of attention that maintains the athlete's focus throughout the competition, i.e., sustained attention, which implies maintaining attentional focus for long periods of time. This is an essential attentional component that prepares the subject to detect unpredictable stimuli over prolonged time periods (*Sarter, Givens & Bruno, 2001*). Considering the extensive literature about the uses and efficiency of the continuous performance task (CPT) (*Smid et al., 2006*), we propose the use of this task in a classical version for the study of sustained attention in martial arts athletes, i.e. sport un-related task in order to avoid advantages related to the sport features

in skilled athletes. Prior to a sustained attention task, subjects are instructed to attend to the same specific target stimulus in the presence of other non-target stimuli. Every stimulus represents a potential target that may require a response.

Considering that combat requires long periods of sustained attention and that this ability should be better developed in skilled athletes, our hypothesis, in accord with previous studies, is that skilled athletes would show a better performance as reflected in larger amplitudes and shorter latencies in the principal ERP components associated with attention, as compared to novice athletes. This difference between groups should also be detectable as prolonged, extended-focus activations in the sustained-attention-related brain regions (i.e., anterior cingulate, prefrontal and parietal areas). We propose that skilled athletes will show better attentional abilities, which will be reflected as better performance in the sustained-attention task, and that this performance can be related to differences in the various components of the ERP (particularly larger amplitudes in the components related to attention: P100, P200 and P300), consistent with previous reports (*Hack, Memmert & Rupp, 2009*; *Hamon & Seri, 1989*; *Hung et al., 2004*; *Jin et al., 2011*; *Ozmerdivenli et al., 2005*; *Zwierko et al., 2011*). Moreover, these behavioral and ERP features will likely correlate with greater activation, in skilled athletes, in the brain structures implicated in sustained attention (anterior cingulate, dorsolateral prefrontal, and parietal cortical regions primarily in the right hemisphere) as previous neuropsychological studies report (*Cohen et al., 1992*; *Fink et al., 1997*; *Pardo, Fox & Raichle, 1991*). Therefore, this study aims to evaluate the differences in sustained attention between skilled and novice martial arts athletes using ERP and sLORETA as tools.

## MATERIALS AND METHODS

### Participants

In order to evaluate sustained attention related to sport expertise in martial arts disciplines, we recruited twenty-one martial arts athletes from judo, tae-kwon-do, and kung-fu disciplines. Degree of combat rank in martial arts is mainly defined by the level of the combat martial training, which includes knowledge and application of the techniques, psychology, and philosophy of the martial arts discipline to real combat. These abilities are assessed with the completion of a theoretical and practical exam and the subsequent delivery of the degree (e.g. a belt with a specific rank). Considering the afore mentioned variables, two groups of athletes were formed: a) 11 skilled athletes (mean age = 25.4 years, SD = 11.5) holding the highest combat degree (i.e. black belt or the highest in each discipline), or/and at least five years of sport practice, a report about the athlete's expertise from the team coach, and competitive experience in national and international competitions. b) 10 novice athletes (mean age = 25.5 years, SD = 9.05) with the lowest combat degree (i.e. no belt or white belt in their discipline), less than one year of sport practice, a report about the athlete's newness from the team coach, and with no competitive experience. All participants were right handed and healthy, with vision that was normal or corrected to normal. All participants showed scores in the normal range (>90) on the Wechsler Intelligence Scale and in the task of variables of attention (TOVA) with a score greater than −1.80, indicating normal attention. No differences in age,

intelligence, ADHD and educational level between groups were observed. Additionally, a mini-mental test (*Folstein, Folstein & McHugh, 1975*) and a neurological evaluation were conducted to confirm no neurological disorders. Before the EEG recording, participants were asked about medication and beverage consumption that could possibly influence the attentional status (coffee and alcohol intake, stimulating drinks, etc.). Athletes who consume or had consumed medications or drugs that affect the nervous system in the last year were eliminated from the study. EEG database of participants from a previous study performed by the authors where motor-related cortical potentials were investigated (*Sanchez-Lopez et al., 2014*) together with data of new participants were analyzed for this paper. Participants were informed of their rights, and they provided written informed consent for participation in the study. This research was carried out ethically and approved by the Ethics Committee of the Instituto de Neurobiología at the Universidad Nacional Autónoma de México. Summary of the characteristics of the participants are detailed in Table 1.

## Stimuli

In this study, the stimuli used were white, 2.95-cm-wide, 2.03-cm-high, arrows pointed in five different directions. The random sequence of arrows was shown at the center of a 17-inch VGA computer monitor on a black background viewed from a distance of 80 cm and at a visual angle of 2.11 × 1.451°.

### Continuous performance task

The task consisted in the presentation of six blocks of 100 arrows each, to complete a sequence of 600 arrows shown to each participant. The subjects were asked to press a button as quickly as possible when the target arrow (pointed right and downward) appeared and not to respond when any other arrow was shown. The stimuli presented were 20% target and 80% non-target. The stimulus duration was 100 ms with an inter-stimulus interval that varied between 1200 and 1500 ms and a response interval overlapped with the inter-stimulus interval (see Fig. 1).

## Procedure

All participants were prepared with the electroencephalographic system and seated in a chair in a dimly lit room. Task instructions asking the subjects to press a button with their right hand as rapidly and accurately as possible when the target stimulus appeared were provided before the CPT performance.

## ERP recording

During the CPT performance, an electroencephalogram (EEG) was recorded using NeuroScan SynAmps amplifiers (Compumedics NeuroScan, Charlotte, NC, USA) and Scan 4.5 software (Compumedics NeuroScan, Charlotte, NC, USA) with 32 Ag/Cl electrodes mounted on an elastic cap. Linked earlobes were used as references. Oculograms were also recorded from a supraorbital electrode, and an electrode was placed at the external canthus of the left eye. A 500-Hz sampling rate was used to digitalize the EEG with a band-pass filter set from 0.1 to 100 Hz. Electrode impedances were maintained below 5 kΩ.

**Table 1 Participants characteristics.**

| Status | Age | Year of sport practice | Intelligence IQ | ADHD score | Sport |
|---|---|---|---|---|---|
| Skilled $N = 11$ | $M = 25.4$ $SD = 11.5$ | $M = 9.4$ $SD = 6.14$ | $M = 102$ $SD = 8$ | $M = 1.79$ $SD = 1.93$ | Judo = 6 TKD = 4 KungFu = 1 |
| Novice $N = 10$ | $M = 25.5$ $SD = 9.05$ | $M = 1$ $SD = 0$ | $M = 107$ $SD = 6$ | $M = 1$ $SD = 1.65$ | Judo = 3 TKD = 2 KungFu = 5 |
| Skilled vs. Novice | NS | $p = .001$** | NS | NS | NS |

Notes:
** $p < .01$.
$M$, mean; $SD$, standard deviation; TKD, tae-kwon-do; NS, No significant differences.

## Continuous Performance Task

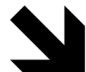

Target = 20%

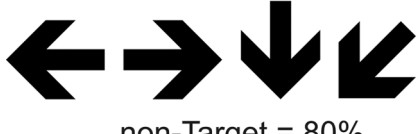

non-Target = 80%

**Figure 1 Continuous Performance Task.** Type, conditions, and probability of stimuli used in CPT.

## Data analysis

### Behavioral analysis

Behavioral analysis was computed using percentages of correct responses, which were transformed [ARCSIN(Square Root (percentage/100))]. Data for the hit rates, false alarms, and response times were compared between groups (skilled and novices) using the two-sample t-test.

### ERP analysis

The ERP were computed offline using 1200 ms epochs from each subject and experimental condition (i.e., target and non-target). Each epoch consisted of the 200 ms preceding the stimulus and the 1000 ms following the presentation of stimulus. Epochs with voltage changes exceeding +80 μV were automatically omitted from the final average. Continuous EEG Segments were visually inspected and those with artifacts and electrical noise were rejected. An eye-movement correction algorithm was applied to remove blinks and vertical ocular-movement artifacts (*Gratton, Coles & Donchin, 1983*). Low pass filtering for 40 Hz and 12 dB slope was performed offline (*Luck, 2005*). Further, a baseline correction was performed using the 200 ms pre-stimulus time window mentioned above. The averaged trials included only those with correct responses.

Statistical analyses of amplitude and latency were separately performed using time windows selected by visual inspection and maximum peak detection to select the time period of all components observed. P100 (100–120 ms), P200 (190–210 ms), N200 (230–290 ms), and P300 (350–500 ms) were the principal waves identified. A series of ANOVAs was also separately performed for each ERP component (time window) and by considering lateral regions or midline electrodes. In order to include the more possible electrodes in the analyses, 24 electrodes from left and right regions were analyzed with Group (skilled and novice) as between-subject factor; Condition (target and non-target), Hemisphere (left and right) and Electrode site (FP1, FP2, F3, F4, C3, C4, P3, P4, O1, O2, F7, F8, T3, T4, T5, T6, CP3, CP4, FC3, FC4, TP7, TP8, FT7 and FT8) as within-subject factors were included. Other series of ANOVAs was performed using midline electrodes. These analyses included Group (skilled and novices) as between-subject factor, and Condition (target and non-target) and Electrode site (FZ, FCZ, CZ, CPZ, PZ, FPZ and OZ) as within-subject factors. The Huynh-Feldt correction was applied to analyses when two or more degrees of freedom in the numerator. Degrees of freedom are reported uncorrected but it is included the epsilon value. The least significant difference (LSD) test was used for *post hoc* multiple pairwise comparisons. Only differences that involved group or any interaction by Group are reported.

### sLORETA analysis

The standardized low-resolution brain electromagnetic tomography (sLORETA) software (http://www.uzh.ch/keyinst/loreta.htm), based on the scalp-recorded electric potential, was used to compute the cortical three-dimensional distribution of current density of the electrophysiological data during the CPT with 32-channel EEG recording, as performed in previous studies (*Perchet et al., 2008*; *Tombini et al., 2009*). The sLORETA method is a three-dimensionally distributed (3D), discrete, linear, minimum norm inverse solution. The sLORETA standardization endows the tomography with the property of exact localization to test point sources, which yields images of standardized current density with exact localization despite its low spatial resolution (i.e., neighboring neuronal sources will be highly correlated). The method has been described in great detail (*Pascual-Marqui, 2002*) and the zero-error localization property is described elsewhere (*Pascual-Marqui, 2007*).

Based on the current sLORETA implementation, computations were made in a realistic head model (*Fuchs et al., 2002*) using the MNI152 template (*Mazziotta et al., 2001*), and with the three-dimensional space solution restricted to cortical gray matter, as established in the probabilistic Talairach atlas (*Lancaster et al., 2000*). The standard electrode positions on the MNI152 scalp were taken from *Jurcak, Tsuzuki & Dan (2007)* & *Oostenveld & Praamstra (2001)*. The intracerebral volume is partitioned into 6239 voxels at a spatial resolution of 5 mm, which allows the generation of images that represent the standardized electric activity at each voxel in neuroanatomic Montreal Neurological Institute (MNI) space. Additionally, images are corrected to Talaraich space and reported using anatomical labels, i.e., Brodmann areas (*Brett, Johnsrude & Owen, 2002*).

To identify differences in current sources between groups each point was analyzed for every component: P100 (between 100 and 120 ms), N150 (between 145 and 165 ms),

P200 (between 190 and 210 ms), N200 (between 230 and 290 ms), and P300 (between 350 and 500 ms). A Statistical Non-Parametric Mapping analysis (10,000 randomizations) was performed with group (skilled and novice) and conditions (target and non-target) as factors. Only the time points where significant differences in the current sources were observed between groups are reported. The following analyses were conducted at these time points: a) an analysis between conditions (target *versus* non-target), separately for each group; b) an analysis between groups in the target condition; and c) an analysis between groups in the non-target condition. Significant differences ($p < .05$) are reported.

## RESULTS

### Behavioral results

Behavioral results showed no significant differences between the two groups of athletes for the rate of correct responses ($t(19) = 0.53$, $p = .60$) or false alarms ($t(19) = -0.22$, $p = .82$). Similarly, there were no differences between groups in response times ($t(19) = 0.41$, $p = .68$) (see Table 2).

### ERP results

P100 and N150 were elicited mainly in occipital areas in both groups when amplitude maps were examined, while P200 was seen in central areas, N200 was distributed in centro-parietal and temporal regions of the left hemisphere, and P300 is observed in the parietal region. Maximum amplitude distribution of the attention effect (i.e. target minus non-target condition) for P100 and P200 seemed to be different between groups: P100 in skilled subjects showed a left-lateral and central distribution while in novice the distribution was mainly lateralized to the right. P200 was observed in centro-parietal for skilled, meanwhile in the novice group this ERP component was centro-frontal. Nevertheless, amplitude differences between conditions in N200 were only observed in novice athletes in the centro-parietal site (see Figs. 2 and 3).

#### P100: 100 to 120 ms time window

Four-way analysis of variance using lateral electrodes data (left and right regions) showed no significant differences between groups (main effect of group $F < 1$), but a significant Condition by Group interaction was found in this time window ($F(1, 19) = 10.75$, $p = .004$). *Post hoc* analyses revealed greater amplitudes elicited by target than non-target condition in the skilled athletes group ($MD = 0.99$ μV, $p = .005$) than in the novice group ($MD = 0.47$ μV, $p = .15$). Additionally, there was a significant Condition by Electrode site by Group interaction ($F(23, 209) = 2.82$, $p = .03$, epsilon = 0.36), where skilled athletes showed greater differences in amplitude between conditions, mainly at P3–P4 ($MD = 1.68$ μV, $p = .006$), O1–O2 ($MD = 1.68$ μV, $p = .001$), T5–T6 ($MD = 1.36$ μV, $p = .008$), CP3–CP4 ($MD = 1.33$ μV, $p = .02$) and TP7–TP8 ($MD = 1.14$ μV, $p = .03$). Three-way ANOVA using midline sites data showed no significant differences between groups (main effect of Group; $F < 1$) or any significant interaction by Group (All $F < 1$).

**Table 2 Behavioral results for CPT.** Rates of hits, false alarms, and response times for both skilled and novice groups.

|  |  | Skilled | Novice |
| --- | --- | --- | --- |
| CPT | Hit rate (%) | *Mean* = 97.2 ± 2.6 | *Mean* = 97.8 ± 1.6 |
|  | False Alarms rate (%) | *Mean* = 0.5 ± 0.3 | *Mean* = 0.6 ± 0.4 |
|  | Response Time (ms) | *Median* = 409.8 ± 48.0 | *Median* = 400.0 ± 59.0 |

Regarding latency analyses, four-way ANOVA with lateral electrodes and three-way ANOVA with midline sites did not display significant main effects of Group (all F < 1) or any interaction by Group (all $F$ < 1).

### P200: 190 to 210 ms time window

Analysis using lateral electrodes showed no significant differences between groups (main effect of Group ($F$ < 1) but does a significant Condition by Group interaction ($F(1, 19) = 9.97$, $p = .005$). *Post hoc* analyses revealed greater differences in amplitude between conditions (target > non-target) in the skilled athletes (MD = 1.39 μV, $p = .008$) compared with the novice group (MD = 0.74 μV, $p = .14$). Three-way ANOVA using midline sites data showed no significant differences between groups (main effect of Group; F < 1) or any significant interaction by Group (All $F$ < 1).

Regarding latency analyses, four-way ANOVA with lateral electrodes and three-way ANOVA with midline sites did not display significant main effects of Group (all F < 1) or any interaction by Group (all $F$ < 1).

### N200: 250 to 300 ms time window

Four-way ANOVA showed no significant main effect of Group ($F(1, 19) = 1.68$, $p = .21$) or any interactions by Group (all $F$ < 1). In contrast, three-way ANOVA of midline sites showed no significant main effect for group ($F(1, 19) = 2.30$, $p = .14$), but there was a significant Condition by Electrode sites by Group interaction ($F(6, 114) = 4.25$, $p = .007$, epsilon = 0.54). *Post hoc* analyses revealed differences between groups in which novices showed higher amplitudes in the target condition than the skilled group at the CZ (*MD* = 5.12 μV, $p = .02$), FCZ (*MD* = 4.27 μV, $p = .04$) and CPZ (*MD* = 5.03 μV, $p = .02$) electrodes. Additionally, differences in amplitude were found between conditions (target > non-target) in novice athletes at the PZ (*MD* = 3.54 μV, $p = .02$) and CPZ (*MD* = 3.92 μV, $p = .009$) electrodes, meanwhile such differences were no observed in skilled athletes.

Regarding latency analyses, four-way ANOVA with lateral electrodes and three-way ANOVA with midline sites did not display significant main effects of Group (all F < 1) or any interaction by Group (all $F$ < 1).

Scalp mean-amplitude maps of P100 and P200 are shown in Fig. 3, where skilled athletes showed higher amplitude than novice athletes. Conversely, higher amplitudes in N200 were observed in novice athletes, and it looks like no differences between groups in their amplitude maps within time window of the P300.

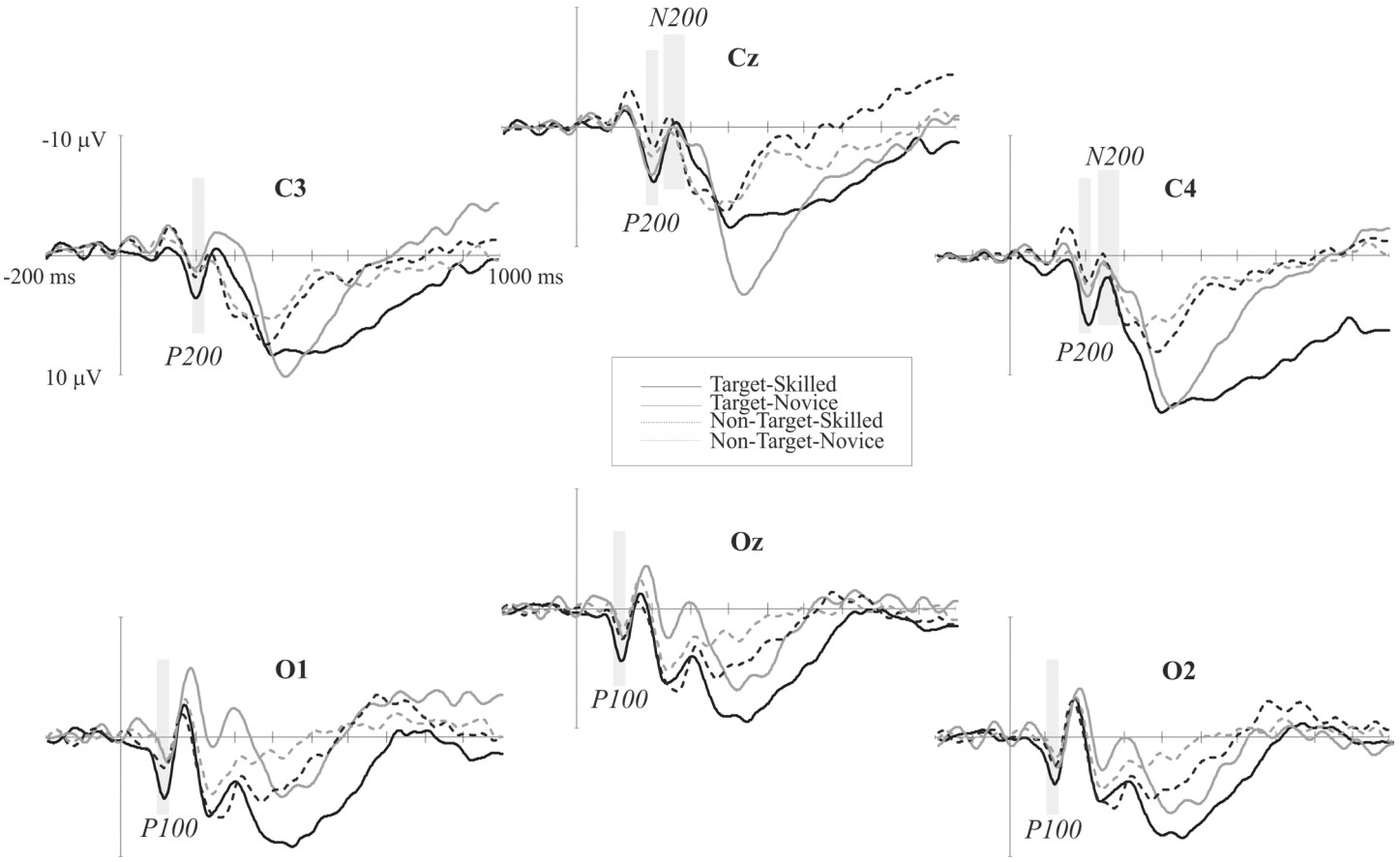

**Figure 2 Event-related potential waves.** ERP grand averages of both target (continuous lines) and non-target (dotted lines) conditions across posterior electrodes. Negative voltage is plotted upward. Black lines represent skilled athletes, and gray lines represent novice athletes. Time windows analyzed in which significant differences were found are shaded gray.

## sLORETA results

Differences in the current sources were found in at least one time point of three ERP components, N150, P200 and P300, when the statistical analysis was conducted with group (skilled and novice) and condition (target and non-target) as factors. No differences in P100 and N200 were observed in the current source analysis. Differences were observed at 146 ms in the Superior Frontal, Medial Frontal, Orbital, Rectal areas of the frontal lobe, and in the Anterior Cingulate of the limbic lobe Gyri, suggesting greater activation in these structures in skilled compared with novice athletes. When the P200 time period was analyzed, greater activation in the Anterior Cingulate of the limbic lobe was observed at 204 ms in skilled than in novice athletes. For the time period of the P300, skilled showed greater activation than novice athletes in different structures at three different latencies: 352 ms (Anterior Cingulate in the limbic lobe and Medial Frontal Gyrus in the frontal lobe), 408 ms (Parahippocampal Gyrus and Sub-Gyral of the limbic lobe, and Fusiform Gyrus in the temporal lobe), and 478 ms (Uncus, Parahippocampal Gyrus and Anterior Cingulate of the limbic lobe, Medial Frontal Gyrus and Superior

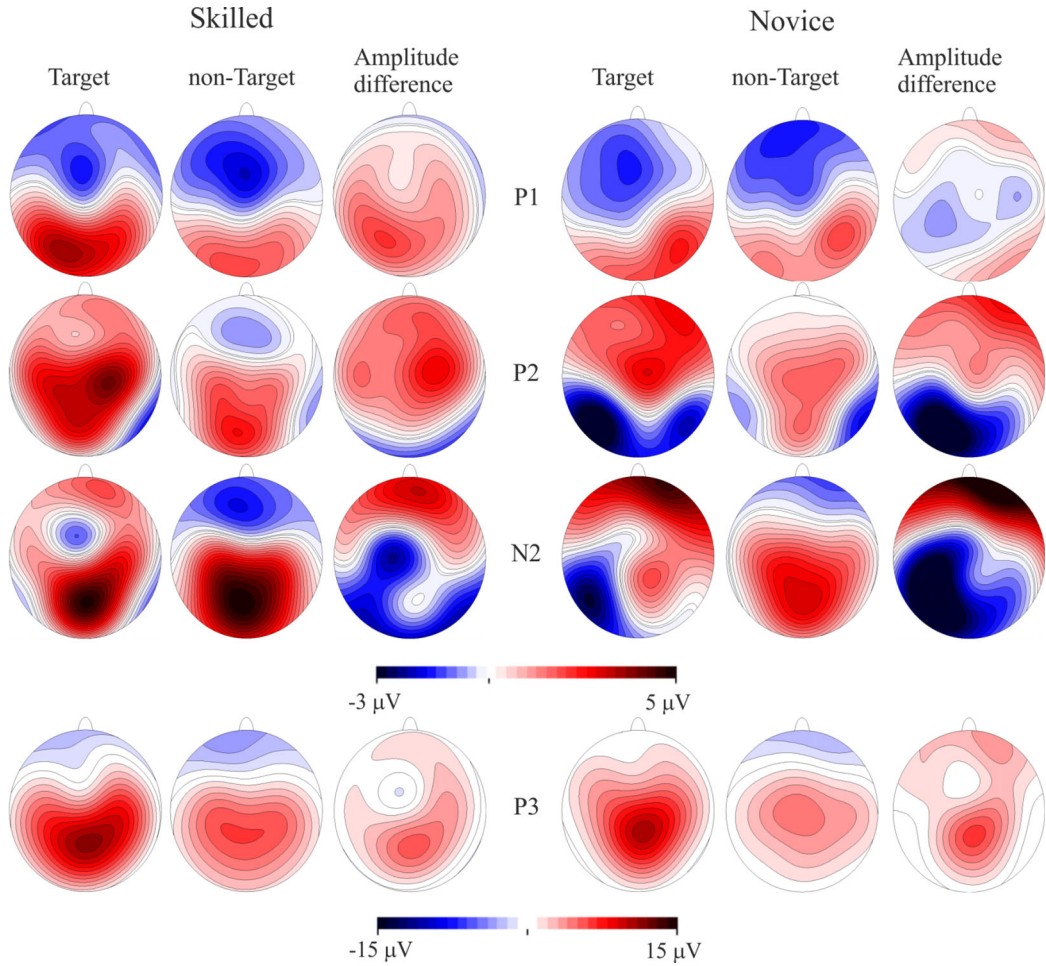

**Figure 3 Event-related potentials topography.** CPT task scalp maps showing representations of the mean amplitudes analyzed in the time windows of the target and non-target conditions and amplitude differences. Skilled athletes are on the left side, and novice athletes are on the right side. Higher P100 and P200 response amplitudes were found in skilled athletes. P300 is also represented, but no significant differences between groups were found.

Frontal Gyrus in the frontal lobe). Detailed results may be seen in Table 3, and the statistical nonparametric maps are shown in Fig. 4.

Where differences between groups were found, the following analyses were conducted: a comparison between conditions (target *versus* non-target), separately for each group; a comparison between groups in the target condition; and a comparison between groups in the non-target condition. Significant differences between conditions were only observed in the skilled group, while significant differences between groups were observed only in the non-target condition. Skilled athletes showed greater activation in the target than non-target condition ($p < .05$) at 146 ms: Precuneus (right), BA 31; Sub-gyral (right), BA 31; and Cingulate Gyrus (right), BA 31; at 352 ms: bilateral Cingulate Gyri, BA 31; bilateral Parahippocampal Gyri, BA 27; Fusiform Gyrus (right), BA 20; bilateral Posterior Cingulate, BA 23; superior Temporal Gyrus (right), BA 23; Insula (right), BA 13; and Sub-gyral (right), BA 21; and 408 ms: Fusiform Gyrus (right), BA 20; and bilateral

**Table 3  sLORETA results for latencies, structures, localization, and statistical values.** Greater differences between conditions were observed in skilled compared to novice athletes.

**Skilled > Novice**

| Latency (ms) | TAL | | | BA | Structure | Cluster | Hemisphere | Value[a] |
|---|---|---|---|---|---|---|---|---|
| | X | Y | Z | | | | | |
| 146 | 25 | 59 | 15 | 10 | Superior Frontal Gyrus (FL) | | Right | 4.55 |
| | 15 | 49 | 7 | 10 | Medial Frontal Gyrus (FL) | 2 | Right | 4.53 |
| | 10 | 44 | 7 | 32 | Anterior Cingulate (LL) | 2 | Right | 4.52 |
| | −5 | 48 | −19 | 11 | Orbital Gyrs (FL) | | Left | 4.45 |
| | −5 | 52 | −24 | 11 | Rectal Gyrus (FL) | | Left | 4.45 |
| | −5 | 52 | −19 | 11 | Superior Frontal Gyrus (FL) | | Left | 4.44 |
| 204 | 5 | 34 | −6 | 32 | Anterior Cingulate (LL) | | Right | 4.43 |
| 352 | 5 | 35 | 12 | 32 | Anterior Cingulate (LL) | 9 | Right | 5.19[b] |
| | 10 | 40 | 16 | 9 | Medial Frontal Gyrus (FL) | | Right | 4.70 |
| | 10 | 34 | 7 | 24 | Anterior Cingulate (LL) | 3 | Right | 4.52 |
| 408 | −15 | −39 | −6 | 30 | Parahippocampal Gyrus (LL) | | Left | 4.81 |
| | −20 | −39 | −6 | 36 | Parahippocampal Gyrus (LL) | 2 | Left | 4.55 |
| | −15 | −44 | −6 | 19 | Sub-Gyral (LL) | | Left | 4.48 |
| | −20 | −44 | −6 | 16 | Parahippocampal Gyrus (LL) | | Left | 4.37 |
| | −25 | −40 | −15 | 37 | Fusiform Gyrus (TL) | | Left | 4.35 |
| 478 | −20 | −11 | −29 | 28 | Uncus (LL) | 7 | Left | 5.29 |
| | −20 | −6 | −29 | 36 | Uncus (LL) | 4 | Left | 5.13 |
| | 0 | 54 | 7 | 10 | Medial Frontal Gyrus (FL) | 38 | Medial | 4.62 |
| | −20 | −11 | −25 | 35 | Parahippocampal Gyrus (LL) | 2 | Left | 4.60 |
| | −5 | 58 | −3 | 10 | Superior Frontal Gyrus (FL) | 6 | Left | 4.39 |
| | 5 | 54 | 16 | 9 | Medial Frontal Gyrus (FL) | | Right | 4.29 |
| | 0 | 48 | −2 | 32 | Anterior Cingulate (LL) | 4 | Medial | 4.26 |
| | −15 | −6 | −21 | 34 | Uncus (LL) | | left | 4.21 |

Notes:
[a] $p < .05$.
[b] $p < .01$.
TAL, Talairach coordinates; BA, Brodmann Area; FL, Frontal Lobe; LL, Limbic Lobe; TL, Temporal Lobe.

Parahippocampal Gyri, BA 36. Differences between groups at 352 ms were observed in the non-target condition ($p < .05$) in the left Inferior Temporal Gyrus (BA 20), left Fusiform Gyrus (BA 20), left Parahippocampal Gyrus (BA 36), left Sub-gyral (BA 20), and left Uncus (BA 20), where novice showed greater activation than skilled athletes.

## DISCUSSION

The goal of this study was to investigate the differences in sustained attention between skilled and novice martial arts athletes using ERP and sLORETA as tools. Based on previous research, we expected to find better behavioral performance, larger amplitudes in attention-related ERP components, and greater activation in the anterior cingulate, dorsolateral prefrontal, and parietal cortical regions, primarily in the right hemisphere brain structures, in skilled than in novice athletes.

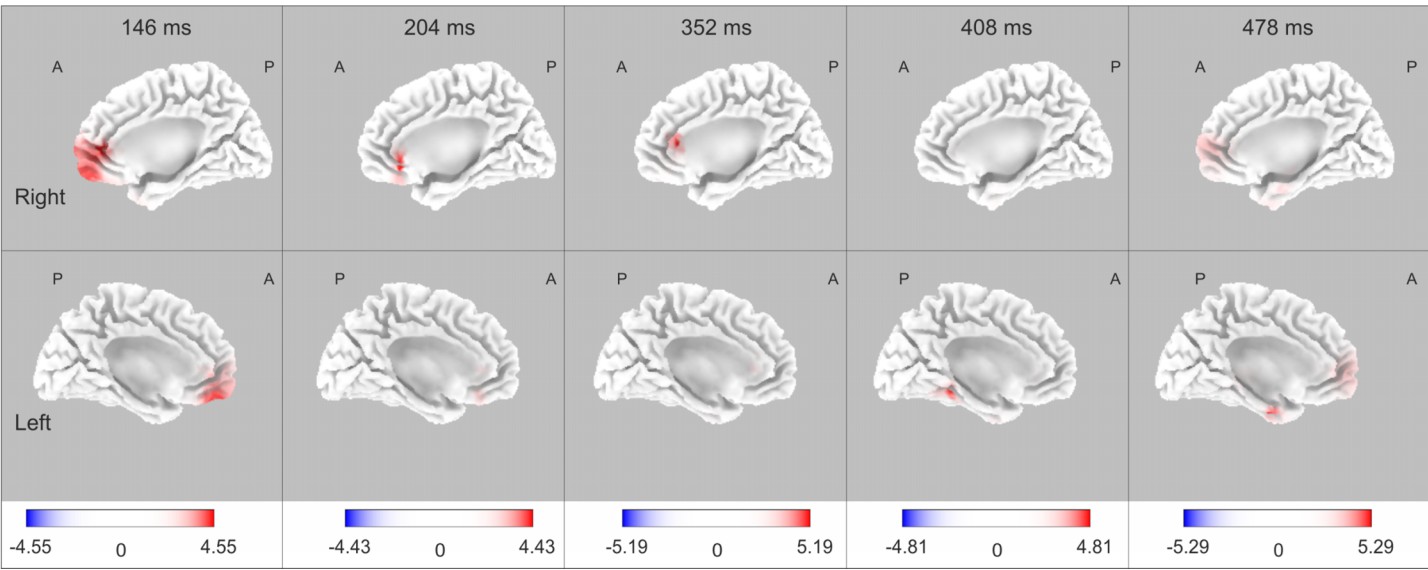

**Figure 4 Current source maps.** Differences are shown at different time points for each component where significant differences were observed: N150 at 146 ms, P200 at 204 ms, and P300 at 352 ms, 408 ms, and 478 ms. Calibration bars indicate *t*-values. Colored areas (red and blue) represent significant values $p < .05$. Positive values mean higher condition differences (target > non-target) in skilled compared with novice athletes, while negative values mean higher condition differences (target > non-target) in novice compared with skilled athletes; the results only showed higher differences in skilled compared to the novice group.

No differences in behavioral results, accuracy or response time, were observed. Since both groups showed high accuracy (almost 100%) and no differences in response time, we can assume that our task was not highly demanding; in fact, most previous studies reported no differences in these variables between expert and less expert athletes or non-athletes. Although no differences in behavioral performance were observed, differences in brain electrical activity were found between groups. These consisted of differences in early ERP components and in activation in the anterior cingulate, frontal, and temporal structures revealed by sLORETA. Our results suggest that: a) ERP and sLORETA seem to be more sensitive tools than behavioral responses to detect differences between groups, and b) there is a different neural pattern for sustained attention in skilled athletes that is likely related to their sport expertise, and these athletes have more efficient neural mechanisms for sustained attention.

Skilled athletes showed significantly greater amplitudes for the target than the non-target stimuli in P100 and P200 than novice athletes, and there were greater amplitudes and differences in amplitude between conditions in novice than skilled athletes for the N200 component. These results suggest differences in early components related to stimulus detection, stimulus evaluation, and decision-making in attentional tasks. On the other hand, the sLORETA results indicated an activation pathway from the frontal to limbic lobe, predominantly to the right hemisphere; this is consistent with the previous reports in sustained-attention tasks that require the basal forebrain cholinergic corticopetal projection system, through direct connections primarily to a right fronto-parietal-thalamic network, for top-down processing such as in the sustained-attention task (*Sarter, Givens & Bruno, 2001*). This pathway was observed more frequently during

the target than during the non-target condition in skilled athletes, which may imply more uniform top-down mechanisms for sustained attention in this group.

The first differences between groups were observed around 100 ms. In the ERP analysis, the larger amplitude in the P100 component in the skilled group could mean a different profile of brain activation modulated by expertise that is associated with spatial attention; this component has been related to the sensitivity of attention to stimulus direction (*Luck, 2005*). In our experiment, participants were instructed to respond to arrows with a particular direction. Although the related literature has not clearly defined this component in relation to the subject skills, our results suggest a greater ability for early detection of the stimulus direction in skilled athletes compared with novices. Previous studies have indicated that acute and habitual exercise affects the early visual-evoked potentials. Neural conductivity in the visual pathway after exercise might be at least partially dependent on the individual's personal training adaptation status (*Ozmerdivenli et al., 2005*; *Zwierko et al., 2011*). These findings could be related to our results, which suggest an adaptation of the P100 component as an effect of training. After this time period, in the range of the N150 component, different structures were involved in the current source analysis. The Superior Frontal Gyrus was the earliest to show a difference between groups; in general, the superior prefrontal area, roughly coinciding with the superior frontal gyrus, is the prefrontal area most consistently activated by sensory stimuli of the three modalities: visual, auditory, and somatosensory (*Fuster, 2008*). An important activation was also observed in the Medial Frontal Gyrus, which is related to fundamental aspects of input-processing streams (*Talati & Hirsch, 2005*). These findings are related to the ERP results and might confirm the suggestion that skilled athletes have an earlier and enhanced ability to detect stimuli.

There were amplitude differences between groups at approximately 200 ms in the target versus non-target comparison. The skilled athletes showed larger differences between conditions than the novices in this posterior P200 component, the nature of which remains unclear in the ERP literature (*Luck, 2005*). Some studies have associated posterior P200 with the initiation of stimulus evaluation and decision-making (*Lindholm & Koriath, 1985*; *Nikolaev et al., 2008*; *Potts, 2004*; *Potts & Tucker, 2001*). The posterior P200 response to an action-anticipation task was different between professional badminton players and non-players, with professionals showing larger amplitudes than non-players; the authors proposed that the players showed superior action-anticipation abilities associated with an enhanced P200 effect that had a posterior-occipital distribution (*Jin et al., 2011*). Additionally, larger amplitudes in the P200 component have been found in sprinters compared with other populations; the authors proposed that smaller amplitudes in the control groups could indicate lower attention levels (*Hamon & Seri, 1989*). Based on these studies and on the hypothesis that the P200 component could be an index of a stimulus-identification process and establishing a perceptual decision (*Lindholm & Koriath, 1985*), this effect in our results likely reflects some generic training effects. The analysis of current sources indicates that the Anterior Cingulate was primarily activated to differentiate between groups. The activation in the Anterior Cingulate was also observed in the different time periods analyzed (corresponding to P100, P200 and

P300 components); a role for the Anterior Cingulate in target detection and executive control has been proposed (*Cabeza & Nyberg, 2000*; *Posner & Petersen, 1990*; *Posner et al., 1988*), and it might also involve the use of information about outcome, particularly reward-related outcome, to guide action selection on the basis of a cost–benefit analysis, integrating information about past action outcomes to optimize voluntary choice behavior (*Bush et al., 2002*; *Hadland et al., 2003*; *Holroyd & Coles, 2002*; *Matsumoto, Suzuki & Tanaka, 2003*; *Rushworth et al., 2004*; *Walton et al., 2006*). These observations are related to our results in the P200 component and the hypothesis that this component is an index of a stimulus-identification process and of establishing a perceptual decision (*Lindholm & Koriath, 1985*), and they confirm that skilled and novice athletes have different neural mechanisms for making perceptual decisions.

Negativity in the central cortical distribution was found at approximately 200 ms. This wave is an N2b (*Naatanen & Picton, 1986*; *Patel & Azzam, 2005*). The N2b corresponds to voluntary processing and is elicited when subjects selectively attend to deviations in oddball paradigms (*Potts et al., 1998*; *Sams, Alho & Naatanen, 1983*). This component, which is typically evoked before the motor response, has been interpreted as a reflection of stimulus identification and distinction (*Patel & Azzam, 2005*), discrimination of a target (*Senkowski & Herrmann, 2002*; *Treisman & Sato, 1990*) and response monitoring (*Stroth et al., 2009*). In visual discrimination tasks, the N2b amplitude is directly correlated with discrimination difficulty (*Senkowski & Herrmann, 2002*). In a previous study that investigated whether exercise and physical fitness have the potential to influence electrophysiological correlates of different aspects of executive control in adolescents using a go/no-go task, the authors found that in higher-fit participants, the N2 amplitude was significantly reduced at the fronto-central electrodes compared with the lower-fit participants; the authors suggested that physical fitness increases the efficiency of the executive control system by reducing the effort required for response-monitoring processes (*Stroth et al., 2009*). Therefore, our results likely point to better executive control in skilled athletes, who do not need to allocate more resources to stimulus discrimination and response monitoring, because the previous stimulus identification and evaluation, and perceptual decision-making were sufficient to provide a motor response.

The afore mentioned differences might be related to differences in the amplitude topography of earlier ERP P100 and P200, which show a slightly different distribution between groups along the scalp. Differences in the scalp topography but not in the source density in the P100 component may be explained as a result of differences in the orientation but not in the location of the dipole that results in differences in amplitude between groups since it is known that amplitude of components also depends on the location and orientation of the dipole (*Mosher et al., 1993*). On the other hand, different sensorimotor mechanisms between skilled and novice athletes allocated in fronto-central and parietal brain cortex would explain differences in topography and current source density in P200. Previous results support a similar argument presented before (*Sanchez-Lopez et al., 2014*), that is to say that a difference in the pointing of the oblique positive dipole gradient can produce topographical differences in the expression of these

components on the scalp as a result of the convergence of sensorial premotor and cognitive processes (*Tomberg et al., 2005*).

No differences between groups in the amplitude of P300 were found. However, a P3b component with parietal distribution was observed in both skilled and novice athletes. This P3b is observed for targets that are infrequent and has been associated with attentional processing (*Luck, 2005*). The results of the current source analysis showed differences at various points along the P300 time period, with activation in frontal, limbic, and temporal structures, i.e., Medial Frontal Gyrus, Superior Frontal Gyrus, Anterior Cingulate, Parahippocampal Gyrus, Sub-Gyral, Uncus, and Fusiform Gyrus. An extensive study investigating multiple brain regions revealed that many cortical areas, including the superior parietal lobe (*Halgren et al., 1995a*), are involved in P300 generation. This is likely the reason why no topographical differences between groups were observed in the amplitude analysis of the P300 recorded on the scalp. Source localization methods, which remove the reference effect, can increase the signal-to-noise ratio, meanwhile amplitude analysis can hide tiny spatial-temporal differences, that are crucial to locate different current sources within conductor volume. Given the foregoing, previous studies have found that the hippocampus and superior temporal sulcus contributed to P3b generation (*Halgren et al., 1995b*). Even frontal brain structures participated in generating P3b: the orbito-frontal cortex, anterior cingulate cortex, and inferior frontal sulcus showed activation during P3 generation (*Baudena et al., 1995*). These findings are closely linked to our results showing an important activation in the target condition in the skilled group that was not observed in the novice group in the P300 time period. This activation might be associated with expertise, as it indicates a different neural pattern for attentional control processes in skilled compared to novice athletes.

In summary, no differences in performance were observed, but differences in amplitude and source analysis of the ERP were found. We propose that brain electrical activity may differentiate between skilled and novice athletes, who adopt different patterns of activity. Indeed, this activity pattern has been observed in previous reports comparing expert and less expert athletes or non-athletes, and it suggests superior cognitive processes in high-level athletes. Two main hypotheses support cognitive superiority of expert athletes in comparison with less expert athletes and non-athletes: The Neural Efficiency hypothesis (for review, see *Babiloni et al., 2009*; *Babiloni et al., 2010a*; *Babiloni et al., 2010b*; *Del Percio et al., 2010*) and the further reinterpretation, Neural Flexibility (for review, see *Spinelli, Di Russo & Pitzalis, 2011*; *Sanchez-Lopez et al., 2014*). The Neural Efficiency hypothesis implies lower spatial cortical activation in expert athletes than less expert groups. In contrast, the Neural Flexibility hypothesis incorporates evidence that some sensorial and cognitive processes increase the recruitment of brain resources in expert athletes. According to source localization analysis results, for the P300 time span, expert athletes showed greater activation than novice athletes across different brain structures. Thus, cognitive superiority of expert athletes seems to be better supported by the Neural Flexibility hypothesis. Although no behavioral advantage of experts means an important discrepancy with this hypothesis, there is a possible explanation regarding some limitations of our experimental task. The time to answer was long enough for expert

athletes to delay their responses to increase their accuracy. Actually, athletes displaying their expertise must include some principles of the discipline philosophy such as controlled-impulse responses. Subjects with high rank in a combat discipline delay their answers to improve their results, i.e. a phenomenon defined by waiting longer and reacting quicker that results in slower but efficient response times (*Vences de Brito & Silva, 2011*). Even though non-significant reaction-time differences were found between groups, the mean reaction-time of experts was higher than that of the novices. For this reason, a lack of behavioral differences should not be considered as evidence against the Neural Flexibility hypothesis.

To our knowledge, few studies have focused on athletes' attention, and none of them has used sLORETA; in fact, some studies have applied source localization analysis to event-related desynchronization data during motor actions and during the judgment of actions (*Babiloni et al., 2009*; *Babiloni et al., 2010a*; *Babiloni et al., 2010b*; *Del Percio et al., 2010*). Our study is the first to clearly define the type of attention studied (i.e., sustained attention) and assess its electrophysiological correlates combining two analyses (ERP and sLORETA), which confirms our hypothesis and demonstrates the effect induced by sport expertise. Additionally, the physiological correlates differentiating these groups, and the electrical activity and brain structures involved in these processes, were characterized, and the results are consistent with the findings of previous studies.

## CONCLUSIONS

Given that skilled athletes showed larger amplitudes in early ERP components than novices, it appears they could detect and make perceptual decisions about the stimulus earlier than novice athletes, an early attention skill that must increase efficiency during combat sports. Current source analysis located brain areas involved in sustained attention, and these areas were consistent with the structures previously found to be activated (*Sarter, Givens & Bruno, 2001*); however, the comparisons between groups showed greater activation, mainly in frontal and limbic lobes that are directly related to sustained attention, in skilled than in novice athletes. This supports the idea that skilled athletes displayed higher attentional abilities than novice athletes. As a whole, this study indicates differences in the neural mechanisms during controlled attention between skilled and novice athletes, likely due to their differing sport expertise.

## ACKNOWLEDGEMENTS

The authors are grateful for the participants' cooperation in this study. The authors also acknowledge Susana A. Castro-Chavira, Leonor Casanova, Lourdes Lara, and Hector Belmont for technical assistance and Dorothy Pless for revising English style.

### Funding

This work was funded by the PROJECT IN205212 from Programa de Apoyo a Proyectos de Investigacion e Innovacion Tecnologica, Direccion General de Asuntos del Personal

Academico and Universidad Nacional Autonoma de Mexico. The funders had no role in study design, data collection and analysis, decision to publish, or preparation of the manuscript.

## Competing Interests

The authors declare that they have no competing interests.

## Author Contributions

- Javier Sanchez-Lopez conceived and designed the experiments, performed the experiments, analyzed the data, contributed reagents/materials/analysis tools, wrote the paper, prepared figures and/or tables.
- Juan Silva-Pereyra analyzed the data, contributed reagents/materials/analysis tools, reviewed drafts of the paper.
- Thalia Fernandez conceived and designed the experiments, contributed reagents/materials/analysis tools, reviewed drafts of the paper.

## Human Ethics

The following information was supplied relating to ethical approvals (i.e., approving body and any reference numbers):

Ethics Committee of the Instituto de Neurobiologia at the Universidad Nacional Autonoma de Mexico.

## Data Deposition

The ERP data is available in the four Supplemental Dataset files.

## Supplemental Information

Supplemental information for this article can be found online at http://dx.doi.org/10.7717/peerj.1614#supplemental-information.

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
