# Peer review of "Sustained attention in skilled and novice martial arts athletes: a study of event-related potentials and current sources"

_PeerJ, doi:10.7717/peerj.1614_

## Round 0.1 · original submission · Major Revisions

Dear authors,

I have received the comments from two reviewers. I fully agree with the reviewers' comments that your manuscript has the potential to be published in PeerJ after you have resolved the issues raised by both reviewers.

With respect and best regards,
Dr Tsung-Min Hung

Reviewer 1 ·

Basic reporting

This study investigated whether sustained attention differs based on the expertise of martial arts disciplines. Sustained attention was assessed using a continuous performance task. The underlying neural mechanisms of sustained attention were reflected by event-related potential (ERP) and current source analysis (LORETA). The authors found that expertise of martial arts sports is associated with superior sustained attention reflected by different patterns of early ERP components and current density during the task. Overall this manuscript was very well-conceived, conducted, and written and will make a significant contribution to the understanding of the association between sustained attention and expertise of martial art sports. However, a few issues need to be addressed before publication.

1.
The eighth paragraph of introduction was a good attempt to demonstrate the specific contribution of the current study by comparing to a previous study conducted by the same authors. However, the entire paragraph may become less relevant as the main focus of the previous study was motor-related process. The authors may either remove this paragraph or integrate it into the literature review in early paragraphs.

2.
The font of note in table 2 should be Times New Roman.

Experimental design

1.
There is a concern regarding the description of the participants. The authors need to add more description of the participants and the quantification of “expertise”. What is the definition of the combat degree in judo, tae-kwon-do, and kung-fu? Are these sports using the same classification? How many subjects were judo athletes? How many subjects were tae-kwon-do athletes? How many subjects were kung-fu athletes? More objective classification of the level of expertise is needed.

Validity of the findings

1.
In the seventh paragraph of introduction, the authors hypothesized that skilled athletes would show a better performance as reflected in larger amplitudes and shorter latencies in the principal ERP components. In the statistical analyses, the author also mentioned “an ANOVA was performed on the mean amplitude and peak latency values to make comparisons between groups across experimental conditions.” However, the results of ERP latencies were not presented in the results section.

2.
What is the rationale to perform the ERP analyses for midline electrodes and lateral electrodes separately? How would the authors explain why the group difference in P1 and P2 amplitudes was selected in lateral electrodes but not midline electrode? In contrast, how would the authors explain why the group difference in N2 amplitude was selected in midline electrodes but not in lateral electrodes?

3.
The lack of RT difference was interpreted as that “martial arts disciplines are not characterized by fast responses, we might not expect shorter reaction times in skilled compared to novice athletes.” This interpretation is not very convincing as fast response is extremely critical in some martial arts sports. Perhaps the task is just too general to reflect the difference caused by expertise of martial arts sports? Perhaps the effect of expertise on RT will be more significant in pre-motor component of RT (i.e. P300 latency)?

4.
In the discussion of P300 amplitude and source analysis along with the P300 time period, it is interesting that many brain regions that are involved in P300 generation exhibited difference between skilled and novice participants but the P300 amplitude measured on the scalp was not differed between two groups. Readers may be curious how would the authors explain this discrepancy

·

Basic reporting

Though I am not English mother tongue and have my own difficulties, I recognize that the paper needs some language editing.

Experimental design

The research design is sound although the sample size is probably a little low and so slightly lacking in power.

Validity of the findings

No comments

Additional comments

Comments to the authors

In this study, it was investigated, with both behavioral and electrophysiological measures, whether skilled martial arts athletes can show difference in sustained attention as compared to novices. Results suggest that athletes have greater task-related modulations of early ERP components, with such effect was more evident in the frontal and limbic lobes, relative to novices. The authors thus concluded that athletes have greater attention allocation during the early stages when performing the Continuous Performance Task.

In general, the rationale for the study is clearly presented and this study can provide additional insights into this area of research. After attentive reading, however, this reviewer has several concerns as indicated below.

Abstract
 In general, the abstract needs to be reformulated.
 The Background part needs re-writing. It is not clear. In fact, the authors should point out the study purpose and its importance here.
 In the Method part the authors should try to briefly state the crucial information with regard to the methods rather than the study purpose.
 It is unclear whether the authors aimed to investigate the ERPs differences in a “sport-related” or “sport-unrelated” task between athletes and non-athletic controls. This should be clarified.
 “When the current density distribution was analyzed…”, needs re-writing

Introduction
 The Introduction is generally well-written. The reviewer don’t find relevant critical points to be addressed, but a need for following amelioration:
 The reviewer suggest that the authors should introduce what kind of cognitive tasks are suitable to investigate sustained attention, and how these task manipulations might possibly relate to the attentional processing in a specific sport (e.g., martial art). Otherwise the readers may loss the logical connection regarding the use of the cognitive task and its application to understand the cognitive processing in athletes.
 Line 74 – 75, “The role of expertise and training in attention and brain activity has been investigated using the ERP technique and sLORETA”, however, only ERP studies are introduced here. Findings with the sLORETA approach should be mentioned as well because this is quite relevant to the present study.
 The reviewer suggest that the authors should provide literature regarding what kind of ERP components may be evoked during the continuous performance task (CPT), this may be useful for the research hypothesis.
 The reviewer suggest that the authors should mention whether the present data was re-analyzed from the previous study (Sanchez-Lopez et al., 2014) investigating motor-related cortical potentials.

Methods
 Line 168 – 173, please provide the duration for the response interval. Does the response interval overlap with the inter-stimulus interval?
 Line 194 – 202, no further off-line filtering for the EEG data was conducted?

Results
 Line 262 – 269, “Electrophysiological data analysis showed significant differences in the P100, P200 and N200 components”, what factor were reported here? Conditions, Group, Electrode, or the interaction? The authors should state it clearly.

Discussion
 The Discussion is generally well-written, this reviewer only have few comments to the data interpretation.
 Line 360 – 361, “these athletes have more efficient neural mechanisms for sustained attention.” The greater amplitude and activation along with no behavioral advantage in athletes found in the present study is inconsistent with the neural efficiency hypothesis in athletes (please see Babiloni et al., 2010; Del Percio et al., 2003). Please provide further interpretation regarding to such discrepancy.

---

## Round 0.2 · accepted · Accept

All comments by the reviewers were addressed to a level of satisfaction. You have my congratulations.